# Soy Protein Isolate–Chitosan Nanoparticle-Stabilized Pickering Emulsions: Stability and In Vitro Digestion for DHA

**DOI:** 10.3390/md21100546

**Published:** 2023-10-22

**Authors:** Pengcheng Zhao, Yuan Ji, Han Yang, Xianghong Meng, Bingjie Liu

**Affiliations:** State Key Laboratory of Marine Food Processing & Safety Control, College of Food Science and Engineering, Ocean University of China, Qingdao 266404, China; zhaopc0320@163.com (P.Z.); yuanji0611@163.com (Y.J.); 15566033503@163.com (H.Y.); mengxh@ouc.edu.cn (X.M.)

**Keywords:** DHA, soy protein isolate–chitosan nanoparticles, Pickering emulsions, stability, oral delivery, cell viability

## Abstract

The purpose of the study was to investigate the stability and oral delivery of DHA-encapsulated Pickering emulsions stabilized by soy protein isolate–chitosan (SPI-CS) nanoparticles (SPI-CS Pickering emulsions) under various conditions and in the simulated gastrointestinal (GIT) model. The stability of DHA was characterized by the retention rate under storage, ionic strength, and thermal conditions. The oral delivery efficiency was characterized by the retention and release rate of DHA in the GIT model and cell viability and uptake in the Caco-2 model. The results showed that the content of DHA was above 90% in various conditions. The retention rate of DHA in Pickering emulsions containing various nanoparticle concentrations (1.5 and 3.5%) decreased to 80%, while passing through the mouth to the stomach, and DHA was released 26% in 1.5% Pickering emulsions, which was faster than that of 3.5% in the small intestine. After digestion, DHA Pickering emulsions proved to be nontoxic and effectively absorbed by cells. These findings helped to develop a novel delivery system for DHA.

## 1. Introduction

Fish oils are rich in polyunsaturated fatty acids, including docosahexaenoic acid (DHA) and eicosapentaenoic acid (EPA). DHA has many physiological functions in humans, such as improving nervous system activity, promoting brain development, restraining obesity, and preventing cardiovascular diseases [1,2]. DHA has attracted attention because of its nutritional functions, and the development and application of DHA-related products have become a hot trend. However, due to its fishy smell, lipophilic structure, and the presence of unsaturated double bonds [3], the application of DHA was limited in common liquids and other food systems. Thus, it is necessary to develop a suitable delivery system to stably distribute it in the system and protect it from oxidation. Encapsulation of lipophilic functional substances using formulations containing lipids has been proven to be an effective method to apply, which can protect them from pro-oxidants present in the surrounding aqueous phase and increase water solubility [4].

The fabrication of delivery systems generally depends on the aim that we want to achieve from them. Generally, there are several factors, as follows: Firstly, the delivery system consists of food-grade ingredients using easy and economical processing operations. Secondly, the system must maintain the physical and chemical stability of bioactive substances during a period of preparation, storage, and digestion. Thirdly, the system has no adverse effect on the physiological environment and sensory properties of humans. Lastly, the ideal system should be the one where the encapsulated bioactive component is released until it reaches a specific site, for example, the small intestine [5,6,7]. The last factor is also the most important because gastrointestinal digestion is a complicated process. In fact, the state of food changes dramatically under a range of physical and chemical (e.g., pH, enzymes, bile salt, etc.) conditions as it passes through the mouth to the stomach and small intestine after digestion. When food enters the mouth, it comes into contact with saliva, experiences heating or cooling to body temperature, changes in ionic strength and pH, and binds to mucin during chewing. After the mouth, food enters the stomach, where it continues to mix with various enzymes (pepsin and gastric lipase) and continues to change in ionic strength and pH [8,9]. Thus, it is crucial to choose a suitable delivery system.

In the food field, emulsions have been applied to encapsulate and deliver bioactive compounds [10]. Traditional emulsions use small molecular weight surfactants and biopolymers as emulsifiers. However, the application of some surfactants may have a negative effect on gut microbiota [11], which limits their application in the food industry. Meanwhile, because of low viscosities, most emulsions are not stable in acidic conditions, leading to fast demulsification in GIT conditions [12]. In recent years, solid-particle-stabilized emulsions (Pickering emulsions) have been widely applied because solid particles could be strongly absorbed into the droplet surface and form a thick shell to maintain the stability of the emulsions. In the food and cosmetic fields, natural ingredients have enhanced consumer interest and improved the bioavailability of targeted substances, which have become the optimal delivery system for functional compounds. Nevertheless, it is difficult to produce large-scale natural edible particles with enough suitability to stabilize emulsions. Thus, the particles need to be modified to be effective stabilizers of Pickering emulsions. Previous kinds of literature suggest that the emulsions stabilized by protein–polysaccharide particles possessed better resistance against coalescence than the emulsions stabilized by protein alone, which is attributed to the greater resistance of the complex compounds [13,14]. Nowadays, many protein–polysaccharide particles are used to stabilize Pickering emulsions, including pea protein-pectin [15], whey protein–chitosan [16], zein–pectin [17], and zein–soybean polysaccharide [18]. These studies focus on preparing Pickering emulsions with various nanoparticles and investigating the physical performance of Pickering emulsions; only a few studies investigate the protection effect and digestion mechanism of Pickering emulsions as the delivery system. Therefore, more research is needed to explore the oral delivery of compounds in Pickering emulsions.

In our previous work, we investigated the preparation of Pickering emulsions stabilized by SPI-CS nanoparticles [19]. The physical stability and rheological properties of Pickering emulsions have also been explored. Up to now, no research has been focused on the protection effect and in vitro digestion of SPI-CS Pickering emulsions for DHA. Hence, in this study, DHA was selected as a targeted lipophilic compound to explore the feasibility of SPI-CS Pickering emulsions as the oral delivery system. The stability of DHA in SPI-CS Pickering emulsions under storage and processing conditions was examined, and the digestion mechanism was illuminated by the simulated GIT model. Caco-2 cells were used to investigate the absorption of DHA after the digestion process.

Prior research has demonstrated that O/W emulsions have specific benefits over bulk oil in terms of lipid oxidation and DHA bioavailability [20]. As a result, the primary focus of our work was on the digestive properties of O/W emulsions containing DHA at various stable nanoparticle concentrations. The aims of this work were to: (1) investigate the stability of DHA under storage, ionic strength, and temperature conditions; (2) measure the retention and release rate of DHA in the in vitro digestion; and (3) examine the toxicity and biocompatibility of the emulsions after the digestion process.

## 2. Results and Discussion

### 2.1. Stability of DHA-Encapsulated Pickering Emulsions

DHA was unstable and susceptible to oxidation because of the presence of many double bonds [3]. The retention rate of DHA during storage for 10 days was shown in Figure 1A. The retention of DHA was 100% for 7 days of storage and decreased to 95% on the tenth day. Although the retention had decreased, DHA-encapsulated Pickering emulsions expressed good stability and a high retention rate. The SPI-CS Pickering emulsions had good protection of DHA, which was attributed to providing screens to resist DHA degradation and oxidation by forming a dense and thick particle layer around oil droplets.

In addition to determining the storage stability of DHA-encapsulated Pickering emulsions, thermal treatment was used in food processing and purchasing. To measure the thermal stability of DHA-encapsulated Pickering emulsions, three temperature modes, including body temperature (37 °C for 4 h) and sterilization temperature (60 °C for 30 min and 90 °C for 3 min), were selected. The results were shown in Figure 1B; the retention of DHA-encapsulated Pickering emulsions was 92–96% under various conditions. These findings reflected that SPI-CS Pickering emulsions used to encapsulate functional substances were stable at various temperatures. According to previous research, octenyl succinic acid (OSA)-modified starch-stabilized Pickering emulsions could be heated to better protect the oil phase because the starch granule stabilized on the oil droplet interface, which allowed the starch to expand and caused a barrier that better covered the oil [21].

Salt is often used in cooking food, and emulsions may be used in products with various concentrations of salt. Thus, it is necessary to determine the ionic strength and stability of DHA-encapsulated Pickering emulsions. The retention rate of DHA was still evaluated as an effect of the NaCl level in Pickering emulsions (Figure 1C). With the increase in ionic strength from 0 mM to 500 mM, the retention was basically maintained at 96%. When ionic strength reached 1000 mM, the retention decreased to 92%. It was attributed to a decrease in electrostatic repulsion between salt and oil droplets due to the electrostatic screening effect. In addition, the retention rate of DHA in all DHA-encapsulated Pickering emulsions with various ionic strengths decreased to 90% after 20 days of storage. Although the concentration of DHA decreased, it remained relatively high in general. These findings were consistent with previous studies [22]. High salt concentrations lowered the stability of lycopene microemulsions due to the decrease in surface charge with an increase in NaCl concentration, which in turn reduced the repulsive force between the droplets.

### 2.2. In Vitro Oral Digestion of DHA-Encapsulated Pickering Emulsions

To explore the mechanism of the in vitro digestion system, DHA-encapsulated Pickering emulsions of different nanoparticle concentrations (1.5 and 3.5%) were prepared to pass through the GIT model. Firstly, the microstructure of DHA-encapsulated Pickering emulsions and the retention rate of DHA were determined in an oral environment. In the mouth stage, there was no significant change in retention compared to the initial emulsions (Figure 2A), which was attributed to DHA-encapsulated Pickering emulsions staying in the mouth for a short time. Figure 3 showed that the size of the emulsion drops as the nanoparticle concentration rises from 1.5% to 3.5%. This might be because droplet agglomeration is inhibited by the high interfacial energy created by the high nanoparticle concentration [23]. Meanwhile, the droplets showed a slight amount of coalescence when DHA-encapsulated Pickering emulsions moved to the simulated mouth stage. These results might be due to depletion and bridging mechanisms in the presence of mucin, which is a large anionic biopolymer in SSF [24] and further induced oil droplets to coalesce and flocculate [25]. Both DHA-encapsulated Pickering emulsions containing different nanoparticle concentrations were stable after exposure to mouth conditions.

### 2.3. In Vitro Gastric Digestion of DHA-Encapsulated Pickering Emulsions

Due to the acidic pH condition and pepsin, the stomach can easily break down various kinds of food. To protect emulsions encapsulated with lipophilic functional active substances from the influence of acidic conditions, it is essential for emulsions to remain stable in the gastric environment [26]. A simulated gastric condition was used to measure the retention rate of DHA and the microstructure of DHA-encapsulated Pickering emulsions containing various nanoparticle concentrations. The retention rate of DHA-encapsulated Pickering emulsions is shown in Figure 2B. The retention rate of DHA decreased to 82% after simulated gastric digestion for 30 min and had no significant change in the next 90 min, which suggested that the DHA-encapsulated SPI-CS Pickering emulsions had good stability at the gastric digestion condition. Compared with the DHA-encapsulated Pickering emulsions containing nanoparticles with a concentration of 3.5% (c = 3.5%), the retention rate of the DHA-encapsulated Pickering emulsions (c = 1.5%) was low at 80% (Figure 2C). The difference in retention might result from two reasons: (1) the SPI-CS Pickering emulsions become slightly unstable at low concentrations [27], and (2) the formation of a gel-like network in SPI-CS Pickering emulsions at c = 3.5% would provide additional protection to resist lipid degradation [28]. In the digestion system, the rate and content of lipid digestion depended on how fast the gel-like network broke down because lipid degradation occurred when the lipases bound to the lipids [29]. The confocal laser scanning microscopy (CLSM) images (Figure 3) suggested that coalescence of the droplets occurred under gastric digestion conditions, which was attributed to the electrostatic screening decreasing the electrostatic repulsion by counter-ions in the simulated gastric fluid (SGF) and the surface charge of the protein-coated droplet changing due to the change from neutral pH to acidic condition [30].

### 2.4. In Vitro Small Intestine Digestion of DHA-Encapsulated Pickering Emulsions

The lipophilic substance was released into the mixed micelle phase during the small intestine digestion, which can be taken as a marker of absorption in the small intestine. Thus, an in vitro small intestine digestion model was used to measure the release rate of DHA and the microstructure of DHA-encapsulated SPI-CS Pickering emulsions (c = 1.5 and 3.5%). The release rate of DHA under the SIF condition was shown in Figure 4. As presented in Figure 4, the release rate of DHA increased to 8% in the SPI-CS Pickering emulsions (c = 1.5%) at the first 2 h, followed by a rapid release of about 26% in the next 1 h. In addition, the DHA-encapsulated Pickering emulsions (c = 3.5%) were observed with a sustained release of about 7% in the whole digestion process. The decreased release rate with increasing nanoparticle concentrations can be attributed to several reasons. First of all, the specific surface area accessible for lipase digestion increases with droplet size, and the larger the interface area that lipase molecules are coated with, the more favorable conditions there are for accelerating the rate of lipid digestion [31,32]. Secondly, SPI-CS nanoparticles formed a protective coating around oil droplets to resist lipase and bile salts entering the droplets. The coating was stable with increasing nanoparticle concentrations. Thirdly, the SPI-CS Pickering emulsions formed a network gel that enhanced the viscosity, inhibiting the movement of lipase to the oil droplets. Previous work had demonstrated that higher nanoparticles had higher viscosity [19]. The DHA-encapsulated Pickering emulsions were released slowly in the process of small intestine digestion, which made them suitable for the body to digest and absorb. The CLSM images showed the change in microstructure in the small intestine digestion (Figure 3). It was shown that the droplet size of the Pickering emulsion with a stable 3.5% particle concentration decreased sharply after the DHA-encapsulated Pickering emulsion was digested in the small intestine, whereas that of the Pickering emulsion with a stable 1.5% particle concentration showed a larger droplet size while most of the droplet size decreased, which may be because there were so few nanoparticles on the droplet surface. Protein in the particles is broken down, which causes the droplets to break and even causes the emulsion to demulsify, releasing lipids [33]. The SPI-CS nanogel-created interface layer prevents lipase and other substances from adhering to the oil droplet’s surface. Instead, bile salt and lipase were only able to absorb and digest lipase by diffusion in the space between the barrier’s layers. DHA release is further inhibited by the high-concentration nanoparticles’ larger interfacial layer, significant steric hindrance, and smaller gap [34]. The reduced concentration of nanoparticles and their digestion in the colon, however, indirectly cause the creation of certain larger-particle-size droplets, which lowers the interfacial spatial barrier of droplets and speeds up the digestion of lipids. The quicker release of DHA at a concentration of 1.5% nanoparticles may also be due to this. Finally, after intestinal digestion, most of the drop size reduction can be explained by several reasons: Firstly, some big oil droplets became small droplets after lipid digestion. Secondly, there were small colloidal particles (insoluble calcium soaps, micelles, and vesicles) after lipid digestion [35]. The DHA-encapsulated SPI-CS Pickering emulsions still observed a few oil droplets during the small intestine digestion, suggesting that the DHA-encapsulated Pickering emulsions could not be completely digested, which might illustrate the delay of lipid digestion [36].

### 2.5. Cell Viability of SPI-CS Pickering Emulsions

To develop a new functional food, it is necessary to determine cell viability to ensure the safety of the food. Cell viability is influenced by several factors, such as composition, chemical environment, surface chemistry, and physical organization [37]. Caco-2 cell lines were treated with SPI-CS Pickering emulsions containing various concentrations of DHA and left untreated as the control group. As shown in Figure 5A, cell viability was basically maintained at 98% at concentrations of 20, 40, 80, 120, and 160 μg/mL and decreased to 92% at the concentration of 200 μg/mL. The cell viability was not affected by any concentration, which indicated no toxicity of DHA-encapsulated SPI-CS Pickering emulsions in cells. In addition, the cells were stained using a Calcein-AM/PI double stain kit (Figure 5B–E), which revealed nearly no dead cells and showed no noticeable dead cell fluorescence. The images also indicated that DHA-encapsulated SPI-CS Pickering emulsions were nontoxic. These findings were consistent with previous studies. SH-SY5Y cells were used to evaluate the biocompatibility of starch nanoparticle stabilization emulsions. The staining distribution of living cells was uniform, and the survival rate of cells was >90%, indicating that the emulsion had no obvious cytotoxicity. The purpose of the preparation of this emulsion is to add it to food as a nutritional fortification agent. Combined with previous studies and dietary guidelines for Chinese residents [38], according to the amount of DHA added in the emulsion, the maximum daily intake of this emulsion is recommended to be 5.74 mL, and the dilution concentration in food should be less than 200 μg/mL.

### 2.6. Cellular Uptake of SPI-CS Pickering Emulsions

Caco-2 cells were used to access the absorption of lipophilic compounds in the small intestine [39,40]. To determine cellular uptake of DHA-encapsulated Pickering emulsions, DHA-encapsulated Pickering emulsions after in vitro digestion were accessed by the addition of Nile red dye into the oil phase. The CLSM images of the control group without DHA-encapsulated Pickering emulsions were shown in Figure 6A,C, which indicate that the cells had no fluorescence. Figure 6B,D showed that the images of SPI-CS Pickering emulsions containing DHA and Nile red were distributed where the cells were located. This result indicated that DHA-encapsulated Pickering emulsions could be uptaken by Caco-2 cells effectively. The findings of the present work were consistent with the previous literature [41]. Magnetic cellulose nanocrystal-stabilized emulsions were stained with Nile red. According to cell uptake images, Nile red and cur were distributed in the cytoplasm of the cells, which proved that the emulsions had cell uptake ability.

## 3. Materials and Methods

### 3.1. Materials

Fish oil (DHA ≥ 70%) and ethyl ester DHA standard were purchased from Qinyecao Biotechnology Co., Ltd. (Xi’an, China). Soy protein isolate (SPI, protein content ≥ 90%) was obtained from Yihaijiali Protein Industry Co., Ltd. (Qinhuangdao, China). Chitosan (MW 3000–6000 Da, degree of deacetylation 90%) was obtained from Hefei Bomei Biotechnology Co., Ltd. (Hefei, China). Corn oil was obtained from Xiwang Food Co., Ltd. (Zouping, China). Simulated saliva fluid (SSF) and simulated intestinal fluid (SIF) were purchased from Shanghai Yuanye Bio-Technology Co., Ltd. (Shanghai, China). N-Hexane was purchased from Acros Co., Ltd. (Beijing, China). Nile red was purchased from Sigma-Aldrich Co., Ltd. (Shanghai, China). 3-(4, 5-Dimethylthiazol-z-yl)-2, 5-diphenyl tetrazolium bromide (MTT), dimethyl sulfoxide (DMSO), and Calcein-AM/PI Double Stain Kit were purchased from Beyotime Biotechnology Co., Ltd. (Shanghai, China). Phosphate-buffered saline (PBS, pH 7.4), dialysis cassettes (molecular weight cut off 3500 Da), and Dulbecco’s modified eagle’s medium (DMEM) were purchased from Solarbio Science & Technology Co., Ltd. (Beijing, China). Fetal bovine serum (FBS) was purchased from Biological Industries Co., Ltd. (Kibbutz Beit Haemek, Israel). Acetic acid, hydrochloric acid, anhydrous ethanol, methanol, NaOH, and NaCl were obtained from Sinopharm Chemical Reagent Co., Ltd. (Beijing, China). Ultrapure water was used for the preparation of all solutions.

### 3.2. Preparation of DHA-Encapsulated Pickering Emulsions

SPI-CS nanoparticles stabilized Pickering emulsions were prepared as described in our previous work [19] The characterization of SPI-CS nanoparticles was shown in Appendix A. DHA was added to the corn oil phase at a concentration of 20%. Briefly, the corn oil phase (φ = 0.5) containing DHA and the SPI-CS nanoparticles (1.5% and 3.5%) were added to a glass bottle. The mixture was homogenized at 15,000 r/min for 2 min with an UltraTurrax^®^T18 (IKA, Berlin, Germany). The DHA-encapsulated SPI-CS Pickering emulsions were stored at 4 °C for further analysis.

### 3.3. Cell Culture

Caco-2 cells were cultured in DMEM, which contained 10% fetal bovine serum, 0.1% nonessential amino acids, 100 μg/mL streptomycin, and 100 U/mL penicillin. The cells were cultured in incubators at 37 °C, 5% CO_2_, and appropriate relative humidity [42].

### 3.4. Microscopy of DHA-Encapsulated Pickering Emulsions

Microscopy images of emulsions were observed from a confocal laser scanning microscope (CLSM) (TCS, SP 8, Leica, Wetzlar, Germany) with 10× objective microscopy, which was used to evaluate structural changes in various conditions. Fresh DHA-encapsulated Pickering emulsions were stained with 0.1% Nile red, which was used to stain the oil phase, and argon laser fluorescence was excited at 488 nm.

### 3.5. Determination of DHA Retention Rate of Pickering Emulsions

#### 3.5.1. Gas Chromatography (GC) Quantification of DHA

The GC system (Agilent 7820, USA) with an FID detector was used to quantify DHA concentration. Column: HP-INNOWax, 30 m × 0.32 mm ID × 0.25 μm (Agilent, Santa Clara, CA, USA) with a detector temperature of 300 °C and injector temperature set at 240 °C. Temperature ramp: starting temperature 170 °C for 5 min, rising 3 °C/min until reaching 210 °C for 30 min. Flame gas: hydrogen 40 mL/min and air 350 mL/min. A 1 mg/mL DHA solution was prepared by adding a 10 mg DHA standard sample and a 10 mL n-Hexane solution. The solution was further diluted to various concentrations (0.04, 0.08, 0.1, 0.2, and 0.4 mg/mL). The DHA concentration of the solution was measured using the above-described GC system and the obtained standard curve.

#### 3.5.2. Extraction and Determination of DHA in Pickering Emulsions

The 1 mL DHA-encapsulated Pickering emulsions were fully mixed with 2 mL of anhydrous ethanol and then 2 mL of n-Hexane solution. The micelle layer was taken out after 10 min, and the other layers were extracted twice with an n-Hexane solution. All micelle layers were combined into a constant final volume of 5 mL. The n-Hexane in the 1 mL micelle layer was blown away with nitrogen, and then 5 mL of hydrochloric-methanol (1:5, *v*/*v*) was added to the test tube. To make sure the DHA was fully methylated, the test tube was then heated in an incubator for 30 min at 70 °C and shaken for 10 s every 5 min. After heating, the tube was cooled to room temperature, and 5 mL of n-Hexane solution was added to take out the micelle phase. The phase was filtered with a 0.22 μm pore-sized filtering membrane and then injected (1 μL) into the GC system. The retention rate was measured as the percent of DHA concentration by Equation (1).
Retention rate (%) = (C_t_/C_0_) × 100%(1)
where C_t_ is the DHA concentration of treated DHA-encapsulated Pickering emulsions, and C_0_ is the DHA concentration of untreated DHA-encapsulated Pickering emulsions.

### 3.6. Stability of DHA-Encapsulated SPI-CS Pickering Emulsions

#### 3.6.1. Storage Stability of DHA-Encapsulated Pickering Emulsions

Freshly prepared DHA-encapsulated SPI-CS Pickering emulsions were stored at 4 °C for 10 days. The storage stability of DHA-encapsulated SPI-CS Pickering emulsions was measured by DHA concentration as a function of storage time. The DHA retention rate was measured at regular intervals by using the method described in previous description.

#### 3.6.2. Thermal Stability of DHA-Encapsulated SPI-CS Pickering Emulsions

The thermal stability of DHA-encapsulated SPI-CS Pickering emulsions was measured under three various thermal conditions [34]. In brief, DHA-encapsulated SPI-CS Pickering emulsions were placed in the tubes and put into a water bath. The DHA retention rate was measured at 37 °C for 4 h, 60 °C for 30 min, and 90 °C for 3 min. The samples were cooled after incubation and measured using the method described above.

#### 3.6.3. Ionic Strength Stability of DHA-Encapsulated SPI-CS Pickering Emulsions

To determine the stability of DHA-encapsulated SPI-CS Pickering emulsions under various ionic strengths, NaCl was added to SPI-CS nanoparticle suspension at various concentrations (0 mM, 100 mM, 300 mM, 500 mM, 1000 mM). Then, the SPI-CS nanoparticle suspension was mixed with corn oil containing DHA and further homogenized as described above. The retention rate of DHA was measured after 1 day and 10 days of storage by using the method described in previous description.

### 3.7. In Vitro Digestion Model

To study the in vitro digestion of DHA, DHA-encapsulated SPI-CS Pickering emulsions were passed through a simulated gastrointestinal (GIT) model, including the mouth, stomach, and small intestinal stages [43]. This model is an international consensus model [44]. For the mouth stage, the SSF was preheated for 2 min at 37 °C before adding the DHA-encapsulated Pickering emulsions. Then, 10 mL of SSF was mixed with 10 mL of initial DHA-encapsulated Pickering emulsions, and the pH of the mixture was adjusted to 6.8. The system was stirred at 100 r/min in the incubator at 37 °C for 10 min. The treated sample was measured for retention rate and visualized microstructures. For the gastric stage, the SGF method was slightly modified from previous work [24] by dissolving 2 g NaCl and 3.2 g pepsin in 1 L of water and adjusting pH to 1.2 using 1 M hydrochloric acid solution. The SGF was preheated for 2 min at 37 °C before mixing, and then 1 mL of sample from the mouth phase (pH 6.8) was mixed with 9 mL of SGF. The mixture was placed in a dialysis cassette, which was placed in 1 L of SGF at 37 °C with continued shaking for 2 h. A total of 1 mL of the treated sample was taken out every 30 min in the total digestion process. The sample was measured for retention rate and visualized microstructures.

For the small intestine stage, the SIF was preheated for 2 min at 37 °C before mixing. A 10 mL sample from the gastric phase (pH 1.2) was mixed with 10 mL of SIF, and the pH was adjusted to 6.8. The mixture was placed in a dialysis cassette, which was placed in the incubator at 37 °C with continued shaking for 3 h. Then, 1 mL of the treated sample was taken out every 1 h during the total digestion process. The sample was measured for retention rate and visualized microstructures.

### 3.8. Cytotoxicity of Pickering Emulsions

Cell viability of digested DHA-encapsulated Pickering emulsions was measured using the MTT method [41]. Caco-2 cell lines were cultured on 96-well plates at a density of 10,000 cells/well and allowed to adhere for 24 h. DHA-encapsulated Pickering emulsions were diluted to various concentrations at 20, 40, 80, 120, 160, and 200 μg/mL and added to each well. Untreated cells were set as the control group. Cell culture fluid was removed after 24 h of treatment, and 10 μL of MTT solution (5 mg/mL MTT) was added to each well. The cells were incubated at 37 °C for 3 h. Then MTT solution was removed and DMSO (100 μL/well) was added to dissolve the formazan product formed in cells. After 30 min, the absorbance at 560 nm was measured using a microplate reader (Biotek, Winooski, VT, USA). The data were measured three times. The percentage of cell viability was calculated by Equation (2).
Cell viability (%) = (N_t_/N_c_) × 100%(2)
where N_t_ is the absorbance of treated cells and N_c_ is the absorbance of untreated cells.

### 3.9. Cell uptake Assay

The cell uptake of DHA in digested SPI-CS Pickering emulsions was visualized following the method described in the literature [42]. DHA-encapsulated Pickering emulsions were stained with 0.1% Nile red to be observed clearly. Caco-2 cell lines were cultured on 48-well plates at a density of 10,000 cells/well. The cells were cultured with Pickering emulsions containing DHA at a concentration of 200 μL/mL for 24 h and normal saline as a control group. The cell culture plate was taken out of the incubator, and 200 μL of PBS (0.01 M, pH 7.4) was added to wash the cells in triplicate to remove the sample. The images of cells were captured using CLSM with 10× objective microscopy using the method described in the previous description.

### 3.10. Statistical Analysis

All experiments were performed in triplicate, and the data were expressed as the mean and standard deviation. The data were analyzed using Origin Pro 2019 software (9.650169) and statistical SPSS Software (Version 25, SPSS Inc., Chicago, IL, USA).

## 4. Conclusions

In this work, Pickering emulsions stabilized by SPI-CS nanoparticles were fabricated to encapsulate and deliver DHA. DHA-encapsulated SPI-CS Pickering emulsions were stable, which showed that the retention rate of DHA still maintained a high level under different temperatures and ion concentrations. DHA-encapsulated SPI-CS Pickering emulsions containing two nanoparticle concentrations (1.5 and 3.5%) were prepared in the pass GIT model. The emulsions had good stability in the mouth and harsh gastric conditions, with only a small loss of DHA. In the small intestine tract, the DHA-encapsulated SPI-CS Pickering emulsions containing low levels of nanoparticles released faster than high levels of nanoparticles. In addition, some oil droplets were not fully digested after the whole digestion process, suggesting that lipid digestion was delayed in the SPI-CS Pickering emulsions. Meanwhile, DHA-encapsulated SPI-CS Pickering emulsions proved to be nontoxic and effectively absorbed by cells. This work might provide insights into the encapsulation and delivery of DHA in SPI-CS Pickering emulsions.

## Figures and Tables

**Figure 1 marinedrugs-21-00546-f001:**
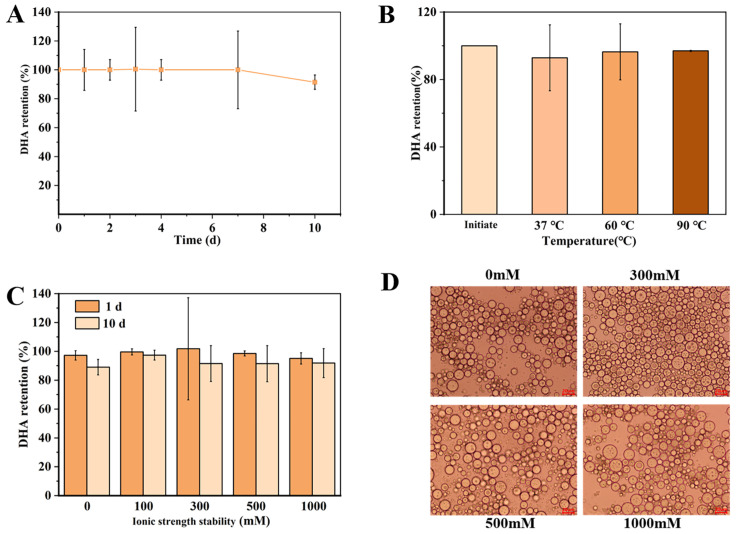
(**A**) The retention of DHA in SPI-CS Pickering emulsions during storage up to 10 days; (**B**) The retention of DHA in SPI-CS Pickering emulsions at various temperatures (37–90 °C); (**C**) The retention of DHA in SPI-CS Pickering emulsions at various ionic strengths (0–1000 mM) after 1 day and 10 days of storage; (**D**) Microscope image of emulsion at different ion concentrations.

**Figure 2 marinedrugs-21-00546-f002:**
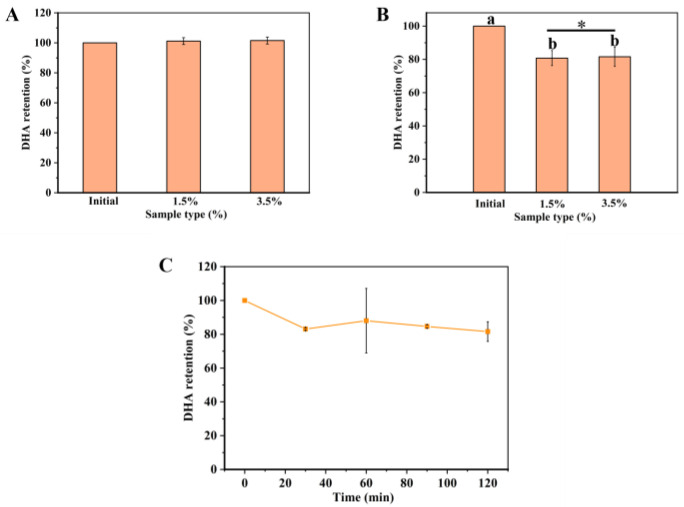
(**A**) The retention of DHA-encapsulated Pickering emulsions containing various concentrations (1.5 and 3.5%) in SSF during storage up to 10 min; (**B**) The retention of DHA-encapsulated Pickering emulsions containing various concentrations (1.5 and 3.5%) in SGF during storage up to 2 h. There were significant differences between the initial group and different concentrations (1.5 and 3.5%) of DHA encapsulation Pickering emulsion with different letters (a, b) (*p* < 0.05).; “*” represents the statistical difference between 1.5% and 3.5% particle concentration data, in general: * (*p* < 0.05); (**C**) The retention of DHA-encapsulated Pickering emulsions (3.5%) in SGF within 120 min.

**Figure 3 marinedrugs-21-00546-f003:**
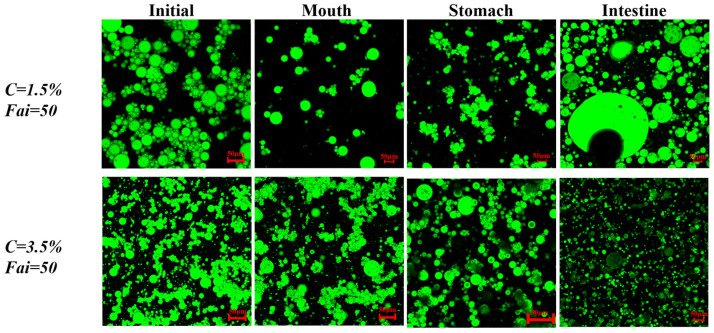
Influence of gastrointestinal tract stage on microstructure (fluorescence microscopy images) of DHA-encapsulated SPI-CS Pickering emulsions with different concentrations (1.5 and 3.5%).

**Figure 4 marinedrugs-21-00546-f004:**
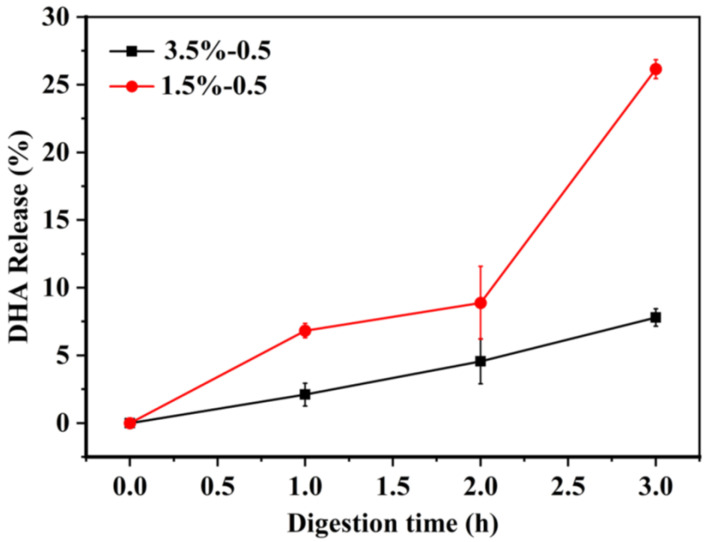
The retention of DHA-encapsulated SPI-CS Pickering emulsions containing various concentrations (1.5 and 3.5%) in SIF within 3 h.

**Figure 5 marinedrugs-21-00546-f005:**
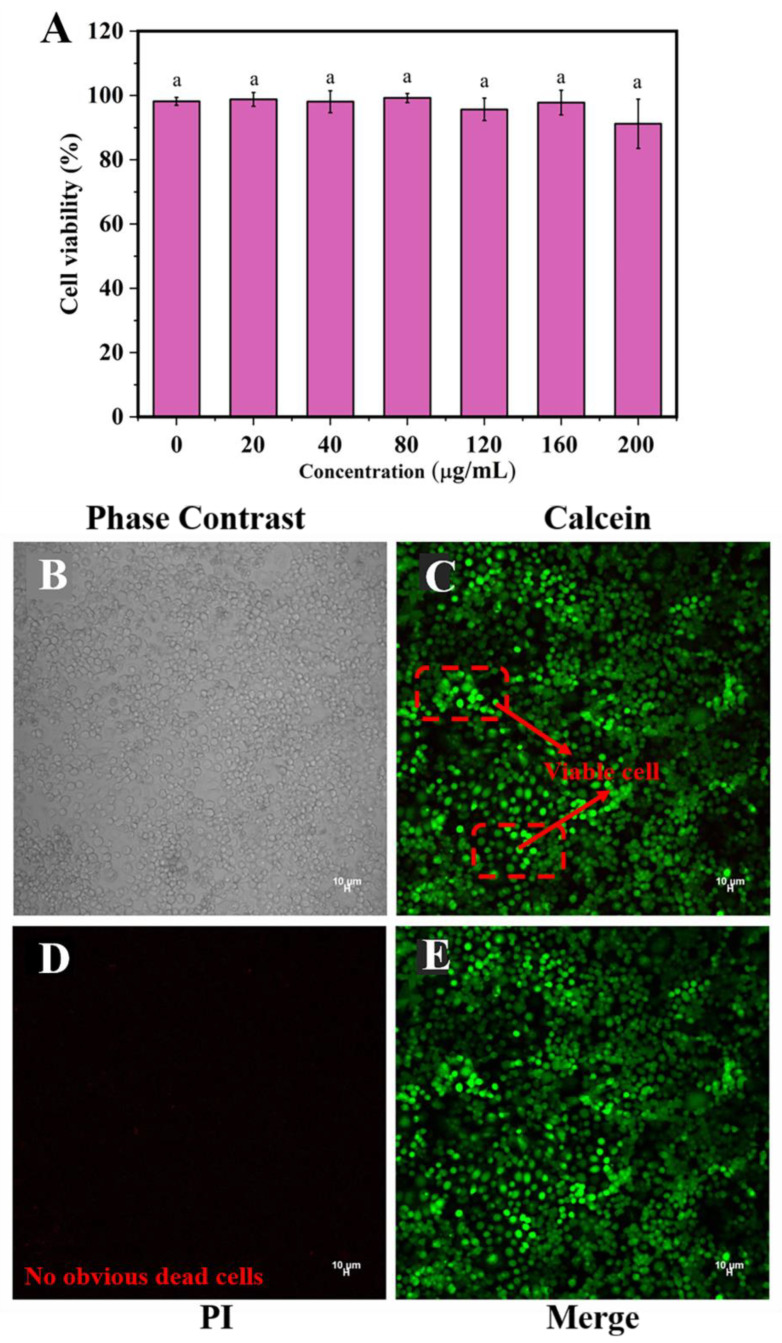
(**A**) Cell viability of SPI-CS Pickering emulsions with various concentrations of DHA (0–200 μg/mL). The same letter “a” in SPI-CS Pickering emulsion treatment groups with different concentrations of DHA (0–200 μg/mL) indicated no significant difference (*p* > 0.05). Fluorescent image of cell viability in SPI-CS Pickering emulsions containing 200 μg/mL DHA: (**B**) phase contrast image; (**C**) the emulsion stained with calcein; (**D**) the emulsion stained with PI; (**E**) the emulsion stained with calcein/PI.

**Figure 6 marinedrugs-21-00546-f006:**
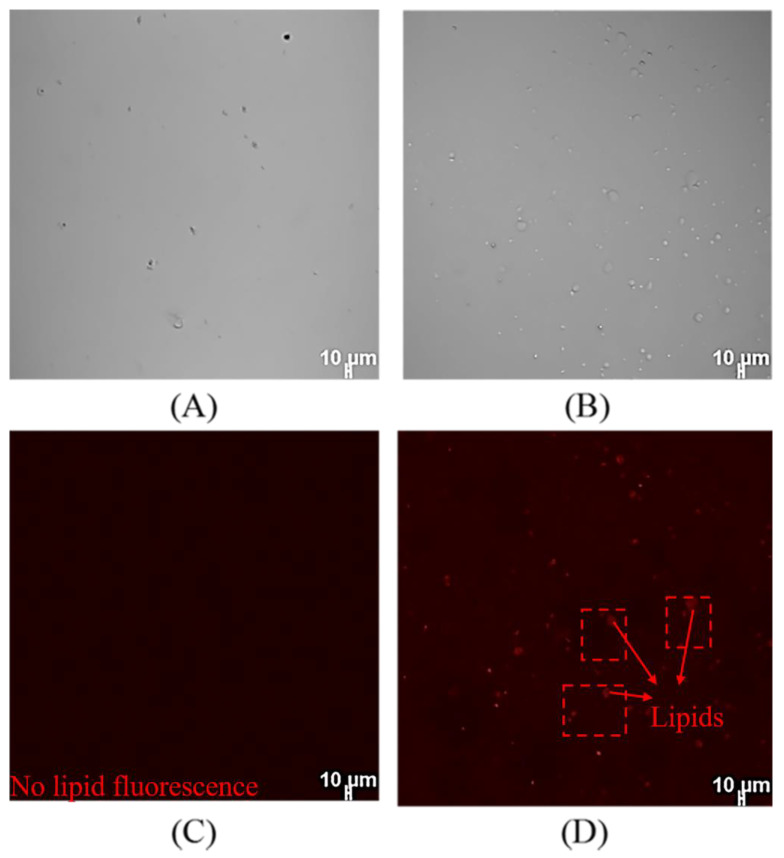
Phase contrast and fluorescent image of cellular uptake of SPI-CS Pickering emulsions containing 200 μg/mL DHA: (**A**,**C**) control normal saline and (**B**,**D**) SPI-CS Pickering emulsions stained with Nile red.

## Data Availability

The data presented in this study are available on request from the corresponding author.

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
