# Peer review of "Soy Protein Isolate–Chitosan Nanoparticle-Stabilized Pickering Emulsions: Stability and In Vitro Digestion for DHA"

_marinedrugs, 2023, doi:10.3390/md21100546_

Round 1
Reviewer 1 Report
The manuscript reports in vitro evaluation of two concentrations of DHA encapsulated Pickering emulsion towards orally available delivery system. I read through the manuscript, but I’m not sure the scientific value of it.
Positive and negative controls are missing in the in vitro assays, for example, DHA itself as a negative control. I don’t understand how significant the emulsion is compared to controls without them.
Regarding cell viability, we can’t argue toxic or non-toxic from the results of the cell viability itself. Because if the effective dose of the emulsion is 200 microg/mL, which is the maximum concentration in the cell viability assay, there is no safety margin between pharmacological effect and toxicity. Therefore, the authors should show the pharmacological effective dose of the emulsion.
Two conditions is not “various conditions”.
Is there any correlation between the size of droplet and the retention of DHA in in vitro assays?
The following literature should be cited because it is closely related in the manuscript.
Food research international, 2022, 162, 112112.
Author Response
Dear Reviewer, Thank you for your letter and for the reviewers' comments concerning our manuscript entitled “Chitosan-soy protein isolate nanoparticles stabilized Pickering emulsion: Stability and in vitro digestion for DHA.”(ID: Marine Drugs-2622648). Those comments are all valuable and very helpful for revising and improving our paper, as well as the important guiding significance to our research, we have studied the comments carefully and have made corrections which we hope meet with approval. Revised portions are marked in red on the paper. The main corrections in the paper and the responses to the comments are as follows:
- Response to comment: Positive and negative controls are missing in the in vitro assays. Response: Regarding your suggestion, we are well aware of the importance of controlled experiments in scientific research. However, the reasons for not using DHA as a negative control in this paper are as follows: This study wanted to verify the intestinal target delivery and sustained release effect of the prepared CS-PIS Pickering emulsion as a carrier of DHA. At this level, we cannot find suitable negative and positive controls at present. However, if it is different from unencapsulated DHA and other carrier forms that can achieve intestinal target delivery and sustained release, such as nanoparticles, microcapsules, nanoemulsions, etc., such analysis is beyond the scope of this paper, and the focus of this paper may be to analyze and compare the differences between different DHA delivery carriers. Therefore, we only compared the two dispersions with stable nanoparticle concentrations that were relatively stable in the previous experiment to study their target delivery and slow-release performance, so as to verify their feasibility as food-grade DHA intestinal target delivery nutritional fortifier.
- Response to comment: authors should show the pharmacologically effective dose of the emulsion. Response: Thank you for your professional advice on the pharmacological level in this article for the toxicity evaluation of the emulsion. We also believe that the suggestion of pharmacologically effective dosage will help to improve the rigor of the toxicity evaluation of the emulsion. However, this paper uses cytotoxicity experiments to conduct a preliminary safety assessment of the emulsion, and the concentration is selected because the concentration of 0-200μg/mL is the appropriate concentration that we consider exists in the food without affecting the quality of the food itself (such as color, odor, etc.). Because the emulsion is prepared with the purpose of serving as a nutritional fortifier added to food, a full evaluation of its toxicology is well beyond the scope of this article, but we have added the following sentence to the cytotoxicity analysis: "The purpose of the preparation of this emulsion is to add it to food as a nutritional fortification agent. Combined with previous studies and dietary guidelines for Chinese residents, according to the amount of DHA added in the emulsion, the maximum daily intake of this emulsion is recommended to be 5.74 mL, and the dilution concentration in food is less than 200 μg/mL."
- Response to comment: Is there any correlation between the size of droplets and the retention of DHA in in vitro assays? Response: We thank the reviewers for their insightful suggestions and agree that it is useful to demonstrate this. Therefore, after consulting the data, we added some analysis in the original text as follows: “It was discovered that the droplet size of Pickering emulsion with stable 3% particle concentration decreased sharply after DHA-encapsulated Pickering emulsion was digested in the small intestine, whereas that of Pickering emulsion with stable 1.5% particles concentration showed larger droplet size while most of the droplet size decreased, which may be because there were so few nanoparticles on the droplet surface. Protein in the particles is broken down, which causes the droplets to break and even causes the emulsion to demulsify, releasing lipids [32]. Second, the SPI-CS nano gel-created inter-face layer prevents lipase and other substances from adhering to the oil droplet's surface. Instead, bile salt and lipase can only absorb and digest lipase by diffusion in the space between the barrier's layers. DHA release is further inhibited by high-concentration nanoparticles' larger interfacial layer, significant steric hindrance, and smaller gap [33]. The reduced concentration of nanoparticles and their digestion in the colon, however, indirectly cause the creation of certain larger particle-size droplets, which lowers the interfacial spatial barrier of droplets and speeds up the digestion of lipids. The quicker release of DHA at a concentration of 1.5% nanoparticles may also be due to this.”
- Response to comment: The following literature should be cited because it is closely related to the manuscript. Food Research International, 2022, 162, 112112. Response: Thanks for the references, which are now included in the revised manuscript Specific references are listed as follows: 31. Wang, J.; Ossemond, J.; Jardin, J.; Briard-Bion, V.; Henry, G.; Le Gouar, Y.; Menard, O.; Le, S.; Madadlou, A.; Dupont, D.; Pedrono, F. Encapsulation of dha oil with heat-denatured whey protein in Pickering emulsion improves its bioaccessibility. Food Res. Int. 2022, 162. Best regards, Bingjie Liu

Reviewer 2 Report
In this paper, the stability of DHA emulsion stabilized by soy protein-chitosan (SPI-cs) nanoparticles and its oral delivery performance were studied. This manuscript surely provides interesting valid data. The following aspects need to be modified by the author.
1. The ordinate unit of Fig1A retention rate should be %, and the entire fig1 figure is too small. What is the reason why the error bars of the 37 ℃ and 60 ℃ data groups in fig1B are so large compared with the other two groups?
2. The vertical coordinates of fig2b are shown unbolded, while groups 2a and 2c are bolded
3. What is the reason for the rapid increase of DHA release rate in Fig3 low concentration 1.5% group? The whole curve presents a very interesting release rule, and the author should add the release data after 3h
4. The fluorescence picture scale of FIG4-5 is too small to see clearly. The fluorescence picture of the small and medium intestine experimental group in fig4 is very interesting. Why c=1.5%, there are large fluorescent particles, shouldn't they be smaller and more easily absorbed
5. fig5A con should be capitalized, ml should be mL, and a stand for no difference?
6. In the cell uptake experiment of Fig6, fluorescence pictures of different time periods should be added, and the fluorescence intensity should be semi-quantitative analysis.
7. The authors should provide size uniformity data of nanoparticles and corresponding electron microscopy images
8. Authors should check the full text for abbreviations that appear when first mentioned, such as CLSM
Author Response
Dear Reviewer,
Thank you for your letter and for the reviewers' comments concerning our manuscript entitled “Chitosan-soy protein isolate nanoparticles stabilized Pickering emulsion: Stability and in vitro digestion for DHA.”(ID: Marine Drugs-2622648). Those comments are all valuable and very helpful for revising and improving our paper, as well as the important guiding significance to our research, we have studied the comments carefully and have made corrections which we hope meet with approval. Revised portions are marked in red on the paper. The main corrections in the paper and the responses to the comments are as follows:
- Response to comment: The ordinate unit of Fig1A retention rate should be %, and the entire fig1 figure is too small. What is the reason why the error bars of the 37 ℃ and 60 ℃ data groups in Fig1B are so large compared with the other two groups?
Response: First, we modified Figure 1, added microscope images of emulsion at different ion concentrations, and enlarged the layout of the entire figure. Secondly, as for the reason why the error bars of the 37℃ and 60℃ data groups in Fig1B are so large compared with the other two groups, we believe that it may be due to the fact that the encapsulation rate of Pickering emulsion for DHA cannot reach 100%, so there will be free DHA in the emulsion system. Free DHA is not completely oxidized and decomposed at this temperature, so it will inevitably be affected by free DHA when sampling, and at 90 degrees Celsius, due to the high temperature, free DHA is oxidized more, and the measurement process is less affected, so there will be this situation.
- Response to comment: The vertical coordinates of fig2b are shown unbolded, while groups 2a and 2c are bolded.
Response: Thank you for checking the details of the picture, we have unified the format of the picture in the original text.
- Response to comment: (1) What is the reason for the rapid increase of DHA release rate in Fig3 low concentration 1.5% group? (2) The whole curve presents a very interesting release rule, and the author should add the release date after 3h.
Response: (1). We believe that the reasons for the faster release of DHA in the Pickering emulsion stabilized by 1.5% nanoparticles in Figure 3 are as follows: 1. A micellar layer is formed on the Pickering emulsion droplet surface as a result of the nanoparticle's attachment to the surface. The SPI-CS nanopillar layer can stop bile salts, etc. from easily absorbing the lipid (DHA) in the droplet. This protective layer's steric hindrance lessens the interface between DHA and other molecules in the intestine, including lipase. They can only make contact with DHA through some of the pores in the nanoparticle micellar layer. The micellar layer on the surface of the droplet is correlated with the concentration of nanoparticles; the lower the concentration, the thinner the protective barrier is created, and the simpler it is to release. 2. Soy protein, which is present in SPI-CS nanoparticles, is likewise broken down by several intestinal proteases. DHA is released when the droplet breaks, the micellar layer thins, and the nanoparticles are destroyed during digestion. At a concentration of 1.5% nanoparticles, its digestion may be sped up. 3. 1.5% The micellar layer may thin in Pickering emulsions with stable nanoparticle concentrations for the reasons mentioned above, and droplets may group together to create larger droplets. A small portion of the droplets in the simulated intestinal fluid seen in Figure 4 enlarge, increasing the distance between the nanoparticles, substances such as bile salt are more likely to absorb DHA in the droplets, Resulting in rapid release of DHA(reason one).
(2) We agree with the reviewers that it would be helpful to elaborate this further with new data. However, we believe that, given the costs involved, expanding our dataset is neither feasible nor would significantly support our case. For this reason, we chose not to make this change, because we have obviously observed the different DHA release behaviors of the two kinds of Pickering emulsion with stable nanoparticle concentration in the intestine within 3h, and analyzed their different release behaviors in combination with Figure 4. Pickering emulsion with 3% nanoparticle concentration is more suitable for intestinal target delivery sustained release of DHA.
- Response to comment: The fluorescence picture scale of FIG4-5 is too small to see clearly. Why c=1.5%, there are large fluorescent particles, shouldn't they be smaller and more easily absorbed?
Response: Your suggestion on the scale of the fluorescence confocal microscopy image is very critical and has been modified in the original draft, regarding your question, the higher the nanoparticle concentration, the smaller the particle size of the stable Pickering emulsion will be at the same water-oil ratio, and vice versa. In this paper, the particle size of the stable Pickering emulsion at c=1.5% is larger than that of the stable Pickering emulsion at c=3%, but the absorption rate is faster. Regarding this, we believe that the factor of release speed should also be taken into account. In the field of delivery, the small size of nanoparticles or emulsion droplets can indeed improve the absorption of content substances, because the small size will have a relatively larger surface area. It can increase the contact area with the intestine. However, when c=1.5%, DHA is released faster in the intestine than 3% (the reasons are explained in detail in the first reply). Therefore, we believe that at the same release rate, the smaller the particle size is, the easier it is to absorb. However, the droplet rupture results in the sudden release of DHA, and the release rate is quite different. The absorption advantage of the small size may not be dominant, resulting in faster absorption of the c=1.5% emulsion.
- Response to comment: The data in Figure 5 were not significant.
Response: First, thank you again for pointing out the error in the picture, we have corrected it in the original. thank you for pointing out the wrong details in FIG. 5. As for your suggestion, we would like to explain that FIG. 5 is about the cytotoxicity determination of DHA Pickering emulsion with different concentrations of Caco-2. Pickering emulsion with different concentrations was added to the cell culture medium for culture, and MTT staining was performed after culture for a period of time. There was no significant difference in cell viability between treated and untreated cells. FIG. 5A showed that endothelin emulsion was not cytotoxic to caco-2 in the 0-200μg/mL concentration range. In Figure 5B-E, cells cultured with Pickering emulsion at a concentration of 200μg/mL were stained to prove the survival of the cells, further verifying the non-toxicity of the emulsion and proving that it can be used as a food-grade nutritional fortifier.
- Response to comment: In the cell uptake experiment of Figure 6, fluorescence pictures of different time periods should be added, and the fluorescence intensity should be a semi-quantitative analysis.
Response: We appreciate the reviewer's insightful suggestion and agree that your suggestion can provide a clearer understanding of the cellular uptake behavior of DHA from Pickering's emulsion, but prior in vitro gastrointestinal simulation could only verify that DHA-coated Pickering's emulsion can release DHA in simulated intestinal fluid. The cellular uptake experiment in Figure 6 can also verify that the emulsion, as a DHA carrier, can not only be released in the intestine but the released DHA can be absorbed by cells, which can provide obvious data and phenomenon support for its targeted delivery in the intestine. However, your suggestion is to further study the relationship between time and intake in the process of DHA uptake by cells. This is beyond the scope of this article, and for that reason, we have chosen not to make this change.
- Response to comment: The authors should provide size uniformity data of nanoparticles and corresponding electron microscopy images.
Response: According to your suggestion, we have added the relevant data to the supplementary data of the original article.
- Response to comment: Authors should check the full text for abbreviations that appear when first mentioned, such as CLSM
Response: Thank you very much for your understanding of the details of the article. We have corrected similar errors in the original text.
We would like to thank the referee again for taking the time to review our manuscript.
Best regards,
Bingjie Liu

Reviewer 3 Report
Some of the data presentation could be improved, for example in Figures 5D and 6C I cannot see anything aside from a black rectangle
Author Response
Dear Reviewer,
Thank you for your letter and for the reviewers' comments concerning our manuscript entitled “Chitosan-soy protein isolate nanoparticles stabilized Pickering emulsion: Stability and in vitro digestion for DHA.”(ID: Marine Drugs-2622648). Those comments are all valuable and very helpful for revising and improving our paper, as well as the important guiding significance to our research, we have studied the comments carefully and have made corrections which we hope meet with approval. Revised portions are marked in red on the paper. The main corrections in the paper and the responses to the comments are as follows:
- Response to comment: Some of the data presentation could be improved, for example in Figures 5D and 6C, I cannot see anything aside from a black rectangle.
Response: We appreciate your thorough comments on the specifics of the images in this article. According to your feedback, we've changed the illustrations in the revised draft, included thorough explanations in Fig. 5D, and changed the brightness and contrast of Fig. 6C to make the dyeing results more obvious. Second, the total black appearance of the images in Fig. 5D and 6C is due to the staining of dead cells in the cytotoxicity test and the staining of intracellular lipid Nile red in the control group (normal saline) in the lipid uptake test in the cell emulsion, respectively, which served as a negative control. A few dead cells are seen in Fig. 5D because of the low toxicity of the emulsion, and the staining outcomes are clear. The absence of Nile red fluorescence in Figure 6C is due to the absence of lipids in normal saline. Therefore, we have also added a description in the article to facilitate readers' understanding, and modified it as follows: Besides, cells were stained using a Calcein-AM/PI double stain kit (Figures 5B, C, D, and E), which revealed nearly no dead cells and showed no noticeable dead cell fluorescence.
Thank you for your valuable comments on this article again!
Best regards,
Bingjie Liu

Round 2
Reviewer 2 Report
The newly submitted manuscript is much improved and provides interesting scientific cases. I suggest publishing this work in Marine Drugs.
Author Response
Dear reviewer, thank you very much for your positive comments.